# Phage integration alters the respiratory strategy of its host

**Jeffrey N Carey[1,2], Erin L Mettert[3], Daniel R Fishman-Engel[4], Manuela Roggiani[4], Patricia J Kiley[3], Mark Goulian[1,4,5]***

[1]Graduate Group in Biochemistry and Molecular Biophysics, Perelman School of Medicine, University of Pennsylvania, Philadelphia, United States; [2]School of Veterinary Medicine, University of Pennsylvania, Philadelphia, United States; [3]Department of Biomolecular Chemistry, School of Medicine and Public Health, University of Wisconsin, Madison, United States; [4]Department of Biology, University of Pennsylvania, Philadelphia, United States; [5]Department of Physics and Astronomy, University of Pennsylvania, Philadelphia, United States

**Abstract** Temperate bacteriophages are viruses that can incorporate their genomes into their bacterial hosts, existing there as prophages that refrain from killing the host cell until induced. Prophages are largely quiescent, but they can alter host phenotype through factors encoded in their genomes (often virulence factors) or by disrupting host genes as a result of integration. Here we describe another mechanism by which a prophage can modulate host phenotype. We show that a temperate phage that integrates in *Escherichia coli* reprograms host regulation of an anaerobic respiratory system, thereby inhibiting a bet hedging strategy. The phage exerts this effect by upregulating a host-encoded signal transduction protein through transcription initiated from a phage-encoded promoter. We further show that this phenomenon occurs not only in a laboratory strain of *E. coli*, but also in a natural isolate that contains a prophage at this site.
DOI: https://doi.org/10.7554/eLife.49081.001

*For correspondence:
goulian@sas.upenn.edu

**Competing interests:** The authors declare that no competing interests exist.

## Introduction

Bacteria and the phages that infect them have a generally antagonistic relationship, with evolution arming each side to defeat the other. Sometimes, though, a bacterium and a temperate phage can form an uneasy truce through lysogeny, wherein the integrated prophage confers some beneficial attribute to its host cell that provides a fitness advantage; after all, unless the host cell dies on the phage's own terms, the phage dies too. Prophage alteration of host phenotype, known as lysogenic conversion (*Lederberg, 1955*), can benefit the host by conferring abilities to produce toxins, resist antibiotics, increase virulence, and repel further phage infections (for recent reviews, see *Argov et al., 2017*; *Bondy-Denomy and Davidson, 2014*; *Davies et al., 2016*; *Fortier and Seku-lovic, 2013*; *Harrison and Brockhurst, 2017*; *Howard-Varona et al., 2017*; *Obeng et al., 2016*; and *Touchon et al., 2017*). Oftentimes these traits are encoded within prophage genetic elements called morons, which contain genes that are regulated by their own promoters and are not involved in the phage lytic cycle (*Hendrix et al., 2000*; *Juhala et al., 2000*). Although lysogenic conversion has been under study since its first description nearly 100 years ago (*Frobisher and Brown, 1927*), there are likely entire classes of phage-encoded proteins that impact host fitness in as-yet-unde-scribed ways, as most phage genes have unknown function and no homology to any genes with known function.

Phages can alter their hosts' behavior in more subtle or indirect ways than, say, carrying a moron that enables toxin production; indeed, in most cases the effects of lysogeny on host physiology are unknown. One study found that deleting all of the cryptic prophages in *Escherichia coli* BW25113

**eLife digest** Animals and plants can all fall prey to viruses – and so can bacteria. The viruses that infect bacteria are called bacteriophages (or phages for short), and they are found everywhere bacteria live and probably outnumber bacteria by at least ten to one.

While some phages quickly kill every bacterial cell they infect, others enter a dormant state by inserting their DNA into the DNA of their host cell. Here they lie in wait for a signal that reactivates them, triggering the production of more phages and the death of the host cell. While the phage lies dormant its DNA may harm the host by interfering with nearby bacterial genes, or it may actually provide new genes that benefit the host. In most cases the effects of dormant phages are unknown.

A bacterium known as *Escherichia coli* is commonly found in the intestines of humans and other mammals. It can use a nutrient called trimethylamine oxide (TMAO) to help it survive rapid decreases in oxygen levels that can occur in its environment. When a phage called HK022 infects *E. coli*, the phage enters a dormant state by inserting its DNA between two genes that are critical for *E. coli* to use TMAO. However, it is not clear what effect, if any, HK022 has on *E. coli*'s behavior.

To address this question, Carey et al. used genetic approaches to study *E. coli* cells carrying dormant HK022 phages. The experiments showed that the bacteria lost the ability to use TMAO to survive rapid decreases in oxygen because the dormant phages switched on one of the neighboring *E. coli* genes. Unexpectedly, the phage achieved this by neatly replacing the gene's own promoter – the stretch of DNA that contains information about when the gene should be switched on, and how strongly – with a substitute promoter carried in the phage's DNA. This substitute promoter is stronger than the normal version – meaning that the gene is more active than it should be.

Phages are key players in every natural population of microbes and are therefore entwined in the health of humans and the environment. The findings of Carey et al. show a new mechanism through which phages modify their hosts. In the future it may be possible to develop this mechanism into a tool to manipulate bacteria in complex environments like infection sites, for example by introducing phages that block the mechanisms that allow bacteria to tolerate antibiotics.
DOI: https://doi.org/10.7554/eLife.49081.002

increased the strain's susceptibility to exogenous stresses and decreased its growth rate through mechanisms yet to be understood (*Wang et al., 2010*). Other studies have shown that host gene expression can be regulated by phage-encoded transcription factors, as in the case of the cI repressor of phage λ that is expressed during lysogeny. This protein prevents expression of the λ lytic genes but was also discovered to act directly at the promoter of the metabolic gene *pckA* (phospho-enolpyruvate carboxykinase), repressing its expression and producing a slow growth phenotype in some conditions (*Chen et al., 2005*). Prophages can also alter host gene expression by means of the position in the host genome where they integrate (*Bondy-Denomy and Davidson, 2014*; *McShan and Ferretti, 2007*). For instance, the Φ13 phage of *Staphylococcus aureus* integrates into the 5' end of the *hlb* gene, disabling β-toxin expression (*Coleman et al., 1991*).

Disruption of host genes by prophage integration can be reversed by prophage excision, and in some bacteria prophages act as switches that regulate host gene expression through controlled excision from interrupted genes, a phenomenon called active lysogeny (*Feiner et al., 2015*). In *Listeria monocytogenes*, for example, the Φ10403S prophage integrates in and disrupts a gene that is required for efficient escape of the mammalian phagosome (*Rabinovich et al., 2012*). The prophage excises during infection, restoring gene function, but its bacterial lysis genes remain repressed. The excised phage later reintegrates back into the same gene without killing its host. In some cases, the prophages involved in active lysogeny have lost the genes required for production of virions but still act as key regulators of cellular processes such as differentiation (*Feiner et al., 2015*).

We became interested in a particular temperate phage that infects *E. coli*—HK022 (*Dhillon and Dhillon, 1972*)—because its integration site lies precisely between the genes *torT* and *torS* (*Yagil et al., 1989*). These genes produce a periplasmic binding protein and a sensor kinase, respectively, that together detect trimethylamine oxide (TMAO) in the periplasm and transduce this signal to the cytoplasm to phosphorylate the response regulator TorR. Phosphorylated TorR then activates transcription of the *torCAD* operon, which encodes TMAO reductase (see *Figure 1A*). This pathway

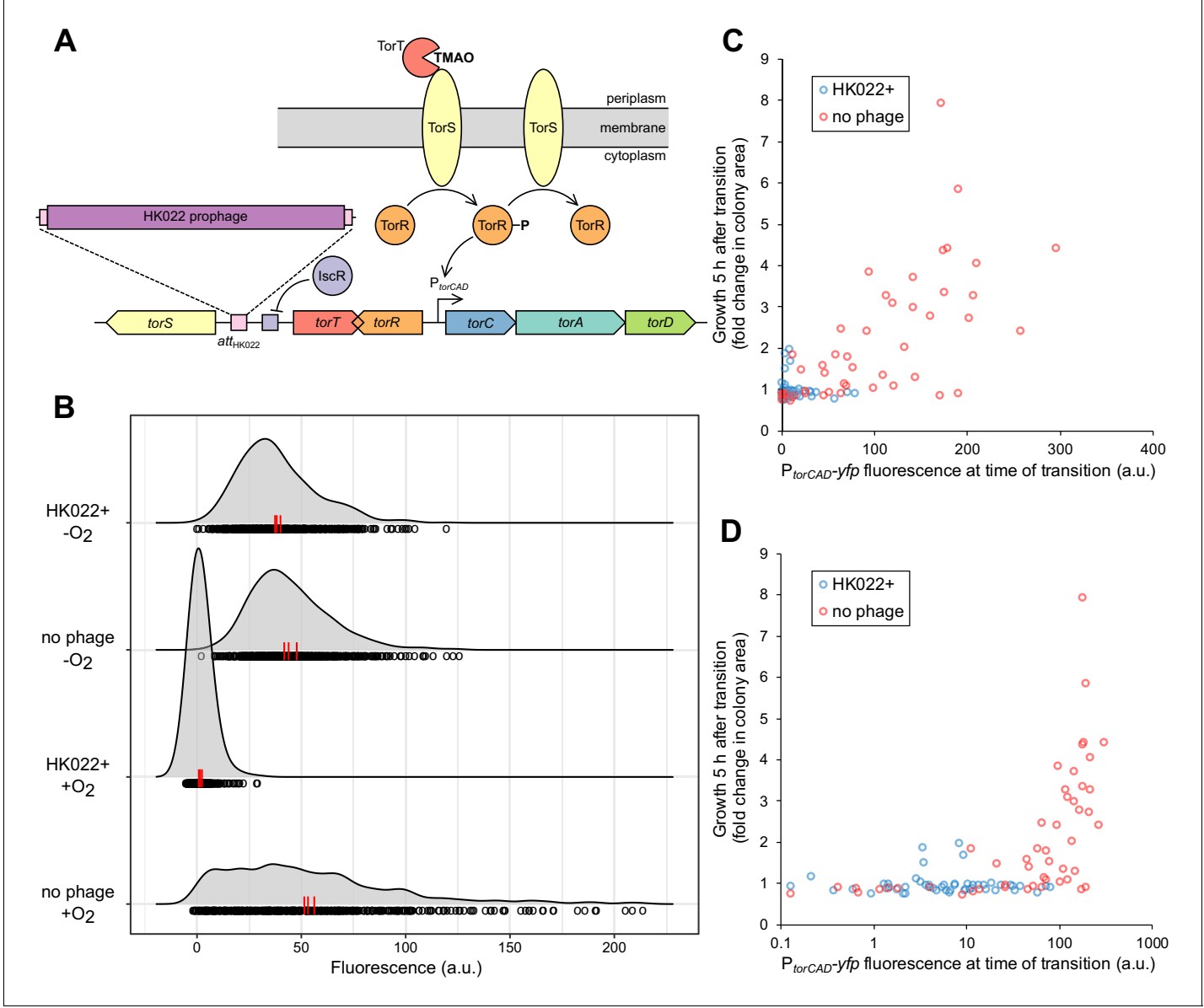

**Figure 1.** Bacteriophage HK022 integrates between the signaling genes *torS* and *torT*, disrupting regulation of *torCAD* and a metabolic bet-hedging strategy. (**A**) HK022 integrates as a prophage at an integration site (*attB*HK022) between *torS* and *torT*, separating *torS* from the IscR binding site that represses its transcription. TorS regulates *torCAD* by phosphorylating and dephosphorylating the transcription factor TorR, which in its phosphorylated state activates transcription from the *torCAD* promoter. To phosphorylate TorR, TorS must interact with TMAO-bound TorT; in the absence of this interaction, TorS dephosphorylates TorR. When oxygen is present, transcription of *torS* and *torT* is repressed to an extremely low level by IscR, and stochasticity in the ratio of TorS to TorT leads to noisy *torCAD* transcription (**Carey et al., 2018**). (**B**) The HK022 prophage shuts off aerobic transcription of *torCAD* but leaves anaerobic expression intact. Distributions of single-cell fluorescence are shown for strains carrying a fluorescent reporter of *torCAD* transcription. Data are shown for an HK022 lysogen (DFE12) and a non-lysogen (MMR8) grown in the presence or absence of oxygen. Each circle represents a fluorescence measurement made in an individual cell. To facilitate qualitative comparisons between distributions, density curves (shown in gray) were generated from single-cell measurements (see Materials and methods). Data are pooled from three independent experiments, with the vertical red lines indicating the population mean fluorescence for each experiment. a.u., arbitrary units. (**C,D**) Most cells carrying the HK022 prophage fail to grow following rapid oxygen depletion. Each circle represents an individual cell monitored for growth following an aerobic-to-anaerobic transition. The same data are presented on a linear scale (**C**) for easier comparison with (**B**) and on a log scale (**D**) for clearer resolution of individual points. The HK022 lysogen (JNC173) constitutively expresses CFP to distinguish it from the non-lysogen (JNC174), which constitutively expresses mCherry. Both strains carry the YFP reporter of *torCAD* transcription and lack *fhuA*, the gene encoding the HK022 receptor. Growth is quantified as the ratio of microcolony area approximately 5 hr after oxygen depletion to the area of the parent cell at the time of depletion. Data are shown for a single representative experiment.

DOI: https://doi.org/10.7554/eLife.49081.003

*Figure 1 continued on next page*

*Figure 1 continued*

The following source data is available for figure 1:

**Source data 1.** Fluorescence measurements for *Figure 1B*.
DOI: https://doi.org/10.7554/eLife.49081.004
**Source data 2.** Fluorescence and growth measurements for *Figure 1C, D*.
DOI: https://doi.org/10.7554/eLife.49081.005

enables *E. coli* to use TMAO as a respiratory electron acceptor. TMAO is widespread in the environment (*Gibb and Hatton, 2004*; *Hatton and Gibb, 1999*) and is particularly abundant in the tissues of many marine organisms (*Eisert et al., 2005*; *Seibel and Walsh, 2002*; *Yancey et al., 1982*). Animals can ingest significant amounts of this compound from seafood-rich diets (*Eisert et al., 2005*; *Zhang et al., 1999*). In addition, humans and other mammals synthesize TMAO from trimethylamine (TMA) that is liberated from dietary precursors by the gut microbiota (*Fennema et al., 2016*; *Zhang et al., 1999*). Circulating TMAO accumulates in urine and is excreted (*Hai et al., 2015*; *Velasquez et al., 2016*).

TMAO respiration allows *E. coli* to grow anaerobically, but it occurs even when oxygen is available (*Ansaldi et al., 2007*). This is surprising because of anaerobic respiration's relatively poor energy yield compared to aerobic respiration. We recently showed that aerobic expression of *torCAD* occurs with high cell-to-cell variability (*Roggiani and Goulian, 2015*), which can benefit the population by serving as a metabolic bet-hedging strategy in the face of a rapid decrease in oxygen availability (*Carey et al., 2018*). Highly variable *torCAD* expression is regulated by oxygen and is mediated by *torS* and *torT*, the genes that flank the HK022 integration site (*Carey et al., 2018*; *Roggiani and Goulian, 2015*). The *torS* and *torT* genes are divergently transcribed but share a repressor binding site for the transcription factor IscR (*Carey et al., 2018*). Under aerobic conditions, IscR repression at this site leads to exceptionally low abundance of TorS and TorT protein and noisy transcription of the *torCAD* operon (*Carey et al., 2018*; *Li et al., 2014*; *Taniguchi et al., 2010*). Curiously, the HK022 integration site separates the *torS* coding sequence from the IscR binding site that regulates its transcription. In this work, we show that the HK022 prophage reprograms the regulation of *torCAD* transcription in *E. coli* by disrupting the native *torS* promoter and introducing a phage-encoded promoter that drives *torS* transcription. By hijacking the regulation of *torS* transcription, HK022 reconfigures how cells respond to the presence of oxygen—in uninfected cells, oxygen regulates cell-to-cell variability in *torCAD* transcription without changing the population mean expression level (*Roggiani and Goulian, 2015*); in infected cells, oxygen regulates the mean *torCAD* expression level and not cell-to-cell variability. Consequently, the HK022 prophage disables the bet-hedging strategy that aids cells during rapid oxygen depletion (*Carey et al., 2018*). We further show that this phenomenon is not unique to HK022 lysogeny in a laboratory strain of *E. coli*, since the *E. coli* isolate NRG 857C, which naturally has a different prophage integrated at the HK022 integration site, shows similar behavior. The mechanism uncovered here, whereby phage cis-acting factors replace those of the host at a particular locus, may be a general mechanism used by temperate phages to alter their hosts' behavior.

## Results

### The HK022 prophage disables aerobic transcription of *torCAD*

The integration site for bacteriophage HK022 is in the short intergenic region between the divergently transcribed genes *torS* and *torT* and separates the *torS* open reading frame from the IscR binding site that negatively regulates *torS* transcription (*Figure 1A*). We suspected that the presence of a prophage at this integration site would disrupt the regulation of *torS* transcription and, ultimately, *torCAD* transcription, which depends on TorS and TorT (*Figure 1A*). To investigate the impact of HK022 lysogeny on *torCAD* transcription, we constructed an HK022 lysogen in an *E. coli* K-12 strain carrying a fluorescent protein reporter of *torCAD* transcription. We grew this lysogenized reporter strain aerobically in the presence of TMAO and measured *torCAD* transcription in single cells by fluorescence microscopy. Transcription of *torCAD* was undetectable in the lysogen but was observed in the non-lysogen control strain (*Figure 1B*).

The simplest explanation for the loss of aerobic *torCAD* transcription in the lysogen is that the presence of the prophage destroys the *torS* promoter, as cells without TorS cannot phosphorylate TorR and activate *torCAD* transcription (*Figure 1A*) (*Jourlin et al., 1996*). Unexpectedly, however, when we measured *torCAD* transcription in cells grown anaerobically in the presence of TMAO, we observed no difference between the lysogen and the non-lysogen (*Figure 1B*). These results indicate that *torS* is still transcribed in the lysogen and that the above explanation is incorrect.

We previously showed that high cell-to-cell variability in aerobic *torCAD* expression can function as a bet-hedging strategy that helps a population tolerate a rapid transition to anaerobiosis (*Carey et al., 2018*). Only cells with a recent history of high *torCAD* expression are able to continue growth after oxygen depletion when TMAO is present and no other respiratory electron acceptors or fermentative substrates are available. Because the HK022 lysogen does not express *torCAD* aerobically, we suspected that it would be unable to employ this bet-hedging strategy and would therefore be unable to grow through an aerobic-to-anaerobic transition under the conditions described above. We tested this hypothesis by growing aerobic liquid cultures of the HK022 lysogen and non-lysogen in media containing TMAO and the non-fermentable carbon source glycerol, combining the cultures, and then transferring to an anaerobic agarose pad, which we used to observe the fates of single cells by time-lapse microscopy. Both strains contained the same fluorescent protein reporter of *torCAD* transcription. Cellular fluorescence was used as a measure of recent *torCAD* transcription and was correlated with cell growth after the transition to anaerobiosis, as in *Carey et al. (2018)*. To differentiate the lysogen from the non-lysogen, each strain was engineered to express a second fluorescent protein constitutively. Both strains carried deletion mutations of the HK022 receptor gene (*fhuA*) to prevent any infection of the non-lysogen by phage particles produced by spontaneous prophage induction in the lysogen. The results of this experiment, shown in *Figure 1C, D*, indicate that only the non-lysogen contains a subpopulation of cells that can grow substantially after oxygen depletion and that this subpopulation has high *torCAD* expression at the time of transition. From this we conclude that the HK022 prophage deactivates TMAO-dependent bet hedging on rapid oxygen depletion.

## The HK022 prophage increases *torS* transcription but not *torT* transcription

Our finding that the HK022 lysogen expresses *torCAD* in the absence of oxygen indicates that the prophage does not simply eradicate the *torS* promoter. To investigate the effect of the prophage on *torS* transcription, we measured β-galactosidase activity produced from an operon fusion of *lacZ* to *torS* in the HK022 lysogen and non-lysogen, both with and without oxygen. We found that *torS* expression was substantially elevated in the HK022 lysogen (*Figure 2A*). When we performed analogous experiments to measure *torT* transcription, we found no difference between the two strains (*Figure 2B*). These results suggest that the HK022 prophage shuts off aerobic *torCAD* transcription not by disrupting *torS* transcription but rather by increasing *torS* transcription while leaving *torT* transcription unchanged. TorS molecules that are not bound to TorT are unable to detect TMAO and are in a state that dephosphorylates TorR (*Figure 1A*). Therefore, cells with a large excess of TorS over TorT would strongly favor TorR dephosphorylation and not express *torCAD* (*Ansaldi et al., 2001*; *Carey et al., 2018*; *Roggiani and Goulian, 2015*). We note that the HK022 lysogen also shows elevated *torS* transcription in the absence of oxygen, and yet *torCAD* is still expressed in these conditions. This suggests that anaerobic TorT levels are sufficiently high for any additional TorS not to have much impact on TorR phosphorylation and *torCAD* expression.

## Increased expression of TorT in an HK022 lysogen restores aerobic *torCAD* transcription

If the model described above is correct, then it should be possible to compensate for elevated TorS levels in an HK022 lysogen and restore aerobic *torCAD* expression by increasing expression of TorT. To test this, we introduced a plasmid containing *torT* under control of a weakened *trc* promoter into the lysogen carrying the fluorescent *torCAD* transcriptional reporter and quantified *torCAD* expression (*Figure 2C*). The result of this experiment agrees with our prediction that the lysogen carrying the *torT* overexpression plasmid is able to express *torCAD* in the presence of oxygen.

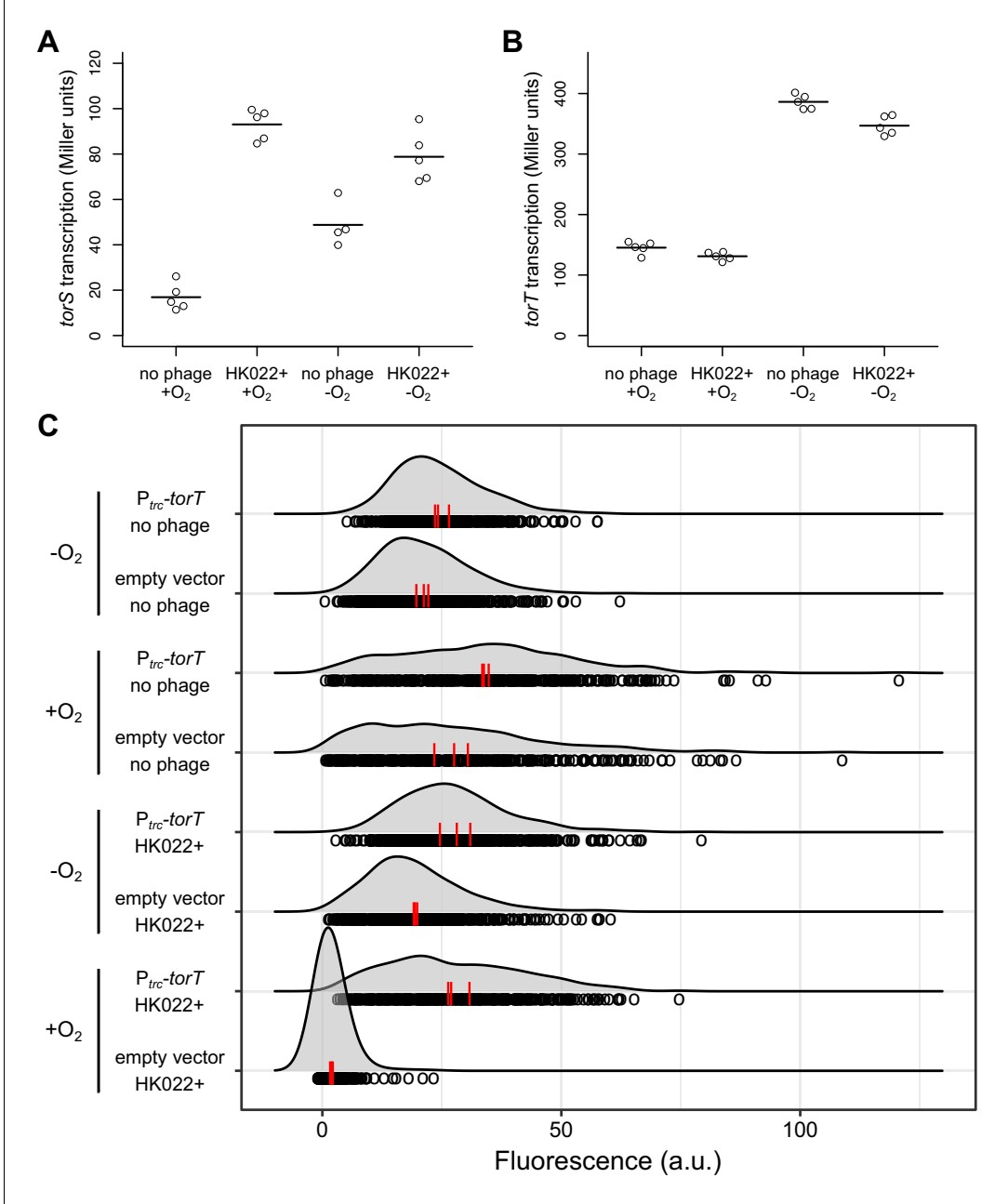

**Figure 2.** The HK022 prophage increases *torS* transcription and has no effect on *torT* transcription. Aerobic and anaerobic transcription of *torS* (A) and *torT* (B) was measured by β-galactosidase assays in strains carrying *torS-lacZ* or *torT-lacZ* operon fusions, with or without the HK022 prophage (strains JNC166, JNC169, JNC163, and JNC168). Each circle represents a measurement obtained from an independent experiment, and the horizontal lines indicate average values. (C) Overexpression of *torT* restores aerobic *torCAD* expression in an HK022 lysogen. The distributions of single-cell fluorescence are shown for strains carrying a fluorescent reporter of *torCAD* transcription. The strains are an HK022 lysogen (DFE12) and a non-lysogen (MMR8) containing a plasmid for *torT* overexpression (pMR26) or an empty vector control (pDSW206), grown in the presence or absence of oxygen. Expression of *torT* from the plasmid is driven by a weakened *trc* promoter without added inducer. Each circle represents a fluorescence measurement made in an individual cell. To facilitate qualitative comparisons between distributions, density curves (shown in gray) were generated from single-cell measurements (see Materials and methods). Data are pooled from three independent experiments, with the vertical red lines indicating the population mean fluorescence for each experiment. a.u., arbitrary units.

DOI: https://doi.org/10.7554/eLife.49081.006

The following source data is available for figure 2:

**Source data 1.** β-Galactosidase measurements for *Figure 2A*.

DOI: https://doi.org/10.7554/eLife.49081.007

*Figure 2 continued on next page*

*Figure 2 continued*

**Source data 2.** β-Galactosidase measurements for *Figure 2B*.
DOI: https://doi.org/10.7554/eLife.49081.008
**Source data 3.** Fluorescence measurements for *Figure 2C*.
DOI: https://doi.org/10.7554/eLife.49081.009

## Transcription of *torS* in a lysogen originates from within the HK022 prophage

We next wanted to probe the mechanism by which the HK022 prophage increases *torS* transcription. We hypothesized that the phage encodes some cis-acting element(s) near its *attP* site that, upon integration, affect *torS* transcription. A simple explanation would be an outward-reading promoter that produces an mRNA transcript originating from within the prophage and reading through *torS*. To test this explanation, we inserted a synthetic terminator construct—an Ω element (*Prentki and Krisch, 1984*)—at the boundary between bacterial and prophage sequence (*attL*$_{HK022}$) (*Figure 3A*). We measured *torS* transcription in strains containing this terminator and found that *torS* expression was very low during both aerobic and anaerobic growth (*Figure 3B*). This result strongly suggests that the mechanism by which the HK022 prophage activates *torS* expression is the production of *torS* transcripts that originate from within the prophage and are driven by a phage-encoded promoter.

The annotated HK022 gene nearest the *torS*-proximal phage/host junction encodes the viral integrase (*int*); there are 73 bp between the *int* stop codon and the junction with the *E. coli* chromosome. In HK022 (as in phage λ), expression of *int* is repressed during lysogeny (*Yagil et al., 1989*), but it is conceivable that transcription of *torS* could be coupled with leaky expression of *int*. To determine whether the HK022 lysogen encodes a separate *torS* promoter, we performed in vitro transcription using DNA sequence upstream of the *torS* start codon. Approximately 200 bp of upstream sequence from the lysogen was cloned into a plasmid, and an analogous plasmid was constructed using upstream sequence from the non-lysogen. In vitro transcription from both plasmids produced transcripts, and the transcripts were different lengths when produced from lysogen sequence than when produced from non-lysogen sequence (*Figure 3C*). Interestingly, transcripts of two distinct lengths were produced from the non-lysogen sequence, suggesting that there are two *torS* promoters when the prophage is absent.

To confirm that the transcripts produced by in vitro transcription were truly *torS* transcripts and to map the transcription start sites associated with each of them, we performed primer extension assays (*Figure 3D, E*). The transcript produced by the HK022 lysogen sequence mapped to a single transcription start site located within the prophage (position indicated in *Figure 3A*). The transcripts produced by the non-lysogen mapped to one transcription start site on the *torS*-proximal side of *attB*$_{HK022}$ and one start site on the *torS*-distal side of *attB*$_{HK022}$ (*Figure 3F*). The transcription start site for the shorter transcript (hereafter TSS$_2$) is identical to a computationally predicted start site (*Huerta and Collado-Vides, 2003*), while the longer transcript (hereafter TSS$_1$) has not previously been predicted or reported. TSS$_2$ is so close to *attB*$_{HK022}$ that its promoter must be at least partially ablated after prophage integration; this likely explains why no TSS$_2$ transcripts are observed in the lysogen. TSS$_1$ transcripts, on the other hand, are likely absent in the lysogen because TSS$_1$ lies on the far side of the *attB*$_{HK022}$ site from the *torS* coding sequence; in the lysogen, the promoter and coding sequence are separated by the entire HK022 genome. It appears, then, that the only *torS* transcripts made in the HK022 lysogen are produced by a phage-encoded *torS* promoter and that the *int* promoter is not required for prophage-regulated *torS* transcription.

After identifying the transcription start sites, we realized that TSS$_2$ is within three base pairs of a predicted translation start site for *torS* and that transcripts produced from TSS$_2$ would not have room for a ribosome binding site. This translation start site uses a GTG start codon, indicated in *Figure 3A, F* (*UniProt Consortium, 2019*). A second translation start site for *torS*, downstream of the GTG start codon and employing an ATG start codon, has also been inferred (*Figure 3A, F*) (*Jourlin et al., 1996*). Neither of these putative start codons is associated with a canonical Shine-Dalgarno sequence, suggesting that translation initiation is inefficient. To determine if either serves as a bona fide start codon, we constructed *lacZ* translational fusions to each and measured β-

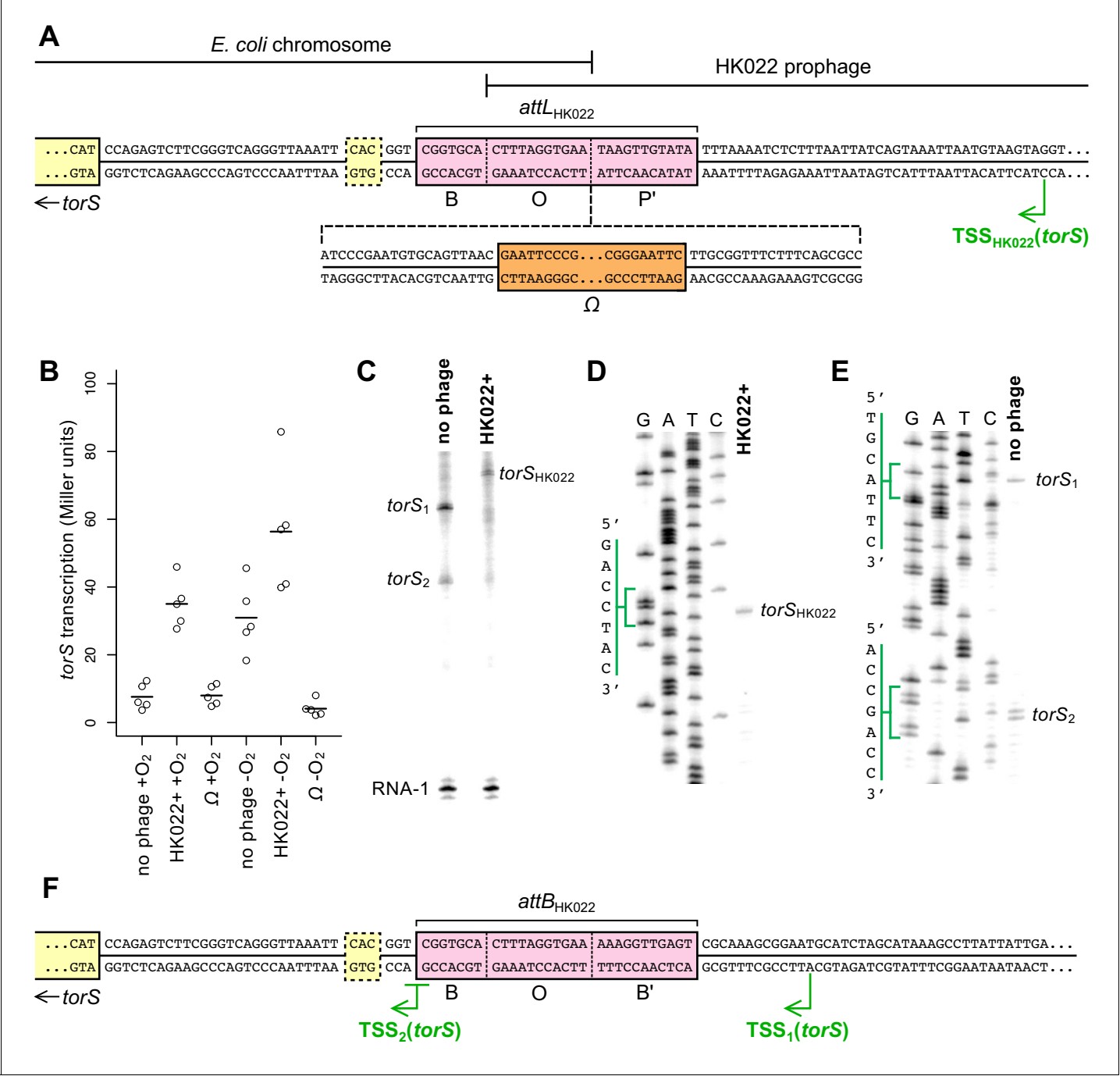

**Figure 3.** Transcription of *torS* in an HK022 lysogen originates from within the prophage. (**A**) Sequence of the torS-adjacent HK022 integration site (*attL*HK022) in an HK022 lysogen. B, O, and P' indicate the bacterial, overlap, and phage segments of the integration site, respectively (*Campbell, 1992*; *Yagil et al., 1989*). The location of the Ω element terminator insertion in strain JNC175 is indicated. In this strain, transcription reading toward *torS* from within the HK022 prophage is blocked by the Ω element. The transcription start site, TSSHK022, was mapped by in vitro transcription and primer extension, shown in (**C**) and (**D**). The previously inferred *torS* GTG start codon is outlined, and the experimentally confirmed ATG start codon is indicated as the start of the *torS* coding sequence. (**B**) Aerobic and anaerobic transcription of *torS* was measured by β-galactosidase assays in strains carrying a *torS-lacZ* operon fusion. Strains contained the wild-type HK022 prophage (JNC169), the prophage with an Ω element (JNC175), or had no prophage at the integration site (JNC166). Each circle represents a measurement obtained from an independent experiment, and the horizontal lines indicate average values. (**C**) In vitro transcription from plasmids containing sequence upstream of *torS* from the HK022 lysogen ('HK022+', pPK13256) or the non-lysogen ('no phage', pPK12669) shows that different transcripts are produced when the prophage is present or absent. Transcription using non-lysogen sequence produces two distinct transcripts, suggesting two transcription start sites for *torS*. RNA-1 is a control transcript for in vitro transcription and gel loading that is generated from a σ70-regulated promoter in pPK13256 or pPK12669. (**D**) Primer extension was performed to map

*Figure 3 continued on next page*

*Figure 3 continued*

the transcription start site of the in vitro 'HK022+' transcript shown in (C). The position of the start site is indicated in (A). (E) Primer extension was performed to map the transcription start sites of the in vitro 'no phage' transcripts shown in (C). The positions of the start sites are indicated in (F). (F) Sequence upstream of *torS* in a non-lysogen, with the *torS* transcription start sites depicted. Transcripts originating from TSS2 can begin at the underlined G or A position, as indicated by the adjacent bands in (E).

DOI: https://doi.org/10.7554/eLife.49081.010

The following source data and figure supplements are available for figure 3:

**Source data 1.** β-Galactosidase measurements for *Figure 3B*.

DOI: https://doi.org/10.7554/eLife.49081.012

**Figure supplement 1.** Identification of the *torS* start codon.

DOI: https://doi.org/10.7554/eLife.49081.011

**Figure supplement 1—source data 1.** β-Galactosidase measurements.

DOI: https://doi.org/10.7554/eLife.49081.013

galactosidase activity. Only the *lacZ* fusion to the downstream ATG produced β-galactosidase activity (*Figure 3—figure supplement 1*). Accordingly, we have indicated the *torS* coding sequence as beginning with the ATG codon in *Figure 3A, F*.

## *E. coli* strains carrying prophages at the HK022 *attB* site are widespread

The above results reveal that the increased *torS* expression caused by the HK022 prophage restricts *torCAD* expression to anaerobic conditions in *E. coli* K-12 strain MG1655, resulting in the loss of bet hedging. This prompted us to investigate the prevalence of prophages integrated at the HK022 integration site in wild *E. coli* strains. We searched for the *torS* and *torT* genes by BLAST (*Boratyn et al., 2013*) against all complete *E. coli* genome sequences available through NCBI (https://www.ncbi.nlm.nih.gov/genome/microbes/) at the time of this analysis and calculated the *torS-torT* intergenic distance for each strain. For all strains with large insertions between *torS* and *torT* (relative to *E. coli* MG1655), we used the PHASTER web server (*Arndt et al., 2016*) to identify prophages. Roughly 5% of sequenced *E. coli* genomes carried prophages integrated immediately upstream of *torS*, and prophage-containing strains were not restricted to closely related *E. coli* phylogenetic groups (*Supplementary file 1*). PHASTER indicated that the prophage integrases were all more similar to HK022 integrase than to any other phage integrase, and a multiple sequence alignment of the genomic region from *torS* to *int* revealed that, for every prophage, the *attL* site was in the same position relative to *torS* as *attL*$_{HK022}$ (*Supplementary file 2*). Sequence conservation was high upstream of TSS$_{HK022}$, suggesting conservation of the associated promoter (despite there being no clearly identifiable −10 or −35 sequences). In most of these strains the *torS-torT* intergenic distance was roughly the same size as the HK022 genome, which is 40,751 bp long (*Juhala et al., 2000*), although several of the strains appeared to have large genomic rearrangements relative to MG1655 in this region. Even in the strains with rearrangements, however, a prophage was integrated immediately upstream of *torS* at *attL*$_{HK022}$.

## Expression of *torCAD* in a prophage-containing wild *E. coli* strain is similar to expression in the HK022 prophage-carrying laboratory strain

We previously showed that *torCAD* expression in various wild *E. coli* strains lacking a prophage between *torS* and *torT* follows a similar pattern to what is seen in MG1655 (*Roggiani and Goulian, 2015*). As prophage integration at *attB*$_{HK022}$ appears to be widespread in wild *E. coli* strains, we wondered whether *torCAD* expression in prophage-containing strains would resemble *torCAD* expression in HK022-infected MG1655. We introduced the fluorescent reporter of *torCAD* expression into one such strain, the Crohn's disease-associated strain NRG 857C (*Eaves-Pyles et al., 2008*; *Nash et al., 2010*). This strain belongs to phylogenetic group B2 (*Supplementary file 1*) and is thus only distantly related to the laboratory strain MG1655 (which belongs to phylogenetic group A). The *torCAD* promoter sequences are identical between NRG 857C and MG1655, enabling us to use the same P$_{torCAD}$-*yfp* reporter construct that we used in MG1655-derived strains to assess *torCAD* transcription in NRG 857C. We integrated the transcriptional reporter into the NRG 857C chromosome by conjugation with an MG1655-derived Hfr donor strain, as NRG 857C is immune to genetic

manipulation by P1 transduction. We used genetic markers in the donor and recipient strains to confirm that the *tor* and *isc* loci of NRG 857C were not replaced upon introduction of the transcriptional reporter (see Materials and methods). When we measured *torCAD* transcription in NRG 857C during aerobic and anaerobic growth by fluorescence microscopy (*Figure 4*), we found that the pattern of expression was much more similar to MG1655 HK022+ than to MG1655 without the prophage (*Figure 1B*). This suggests that in at least some wild *E. coli* strains there is prophage-mediated regulation of *torCAD* expression that is mechanistically similar to the HK022-mediated regulation seen in MG1655.

## Discussion

In this work, we have shown that bacteriophage HK022 reconfigures the regulation of TMAO reductase expression in *E. coli*. Although other cases have been described wherein a prophage alters the expression of host metabolic genes, we are unaware of other instances in which a prophage so dramatically modifies its host's response to the presence of a metabolite. By restricting *torCAD* expression to anaerobic conditions, HK022 converts oxygen-dependent regulation of the variance in *torCAD* expression (seen in non-lysogens) into oxygen-dependent regulation of mean *torCAD* expression (*Figure 5*).

HK022 reconfigures *torCAD* regulation by increasing expression of the regulatory protein TorS (*Figure 5*). The phage appears to achieve this by replacing the native *torS* promoters with a promoter located within the prophage. We mapped the transcription start site associated with this promoter to within the prophage and were able to abolish *torS* transcription initiated at this site by inserting a transcriptional terminator into the junction between the prophage and the *E. coli*

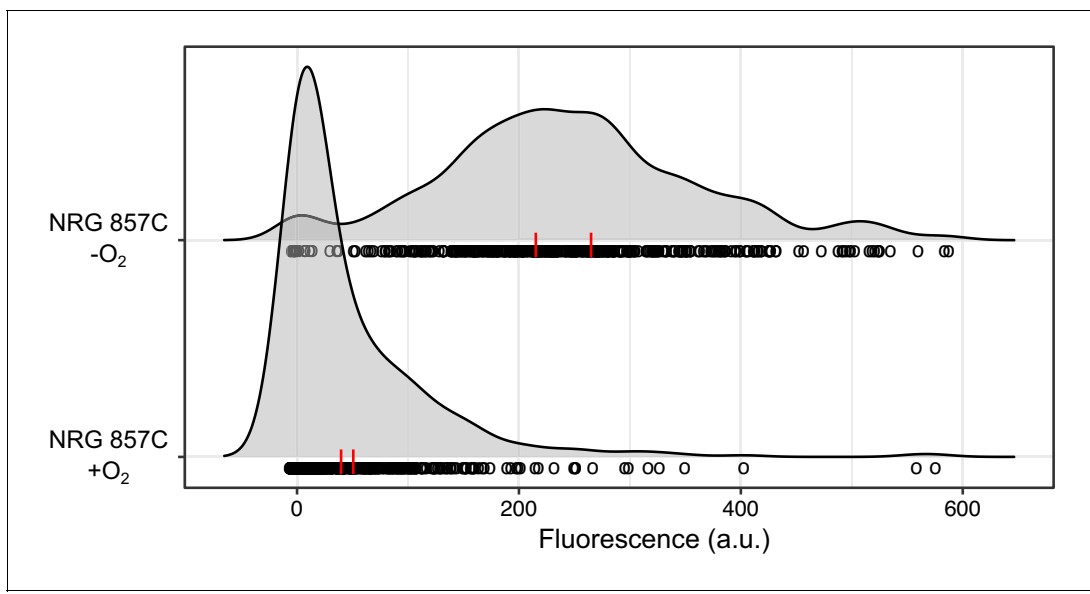

**Figure 4.** A wild *E. coli* strain carrying a prophage at *attB*HK022 shows an oxygen-dependent *torCAD* expression pattern similar to that of the HK022-infected laboratory strain. A fluorescent reporter of *torCAD* transcription was introduced into the Crohn's disease-associated *E. coli* strain NRG 857C and used to measure expression during aerobic and anaerobic growth. NRG 857C naturally carries a prophage at the HK022 integration site and displays a qualitatively similar pattern of *torCAD* expression as HK022-infected MG1655 (see *Figure 1B*). Distributions of single-cell fluorescence are shown for the NRG 857C P*torCAD*-*yfp* strain (DFE34), with each circle representing a fluorescence measurement made in an individual cell. To facilitate qualitative comparisons between distributions, density curves (shown in gray) were generated from single-cell measurements (see Materials and methods). Data are pooled from two independent experiments, with the vertical red lines indicating the population mean fluorescence for each experiment. a.u., arbitrary units.
DOI: https://doi.org/10.7554/eLife.49081.014

The following source data is available for figure 4:

**Source data 1.** Fluorescence measurements.
DOI: https://doi.org/10.7554/eLife.49081.015

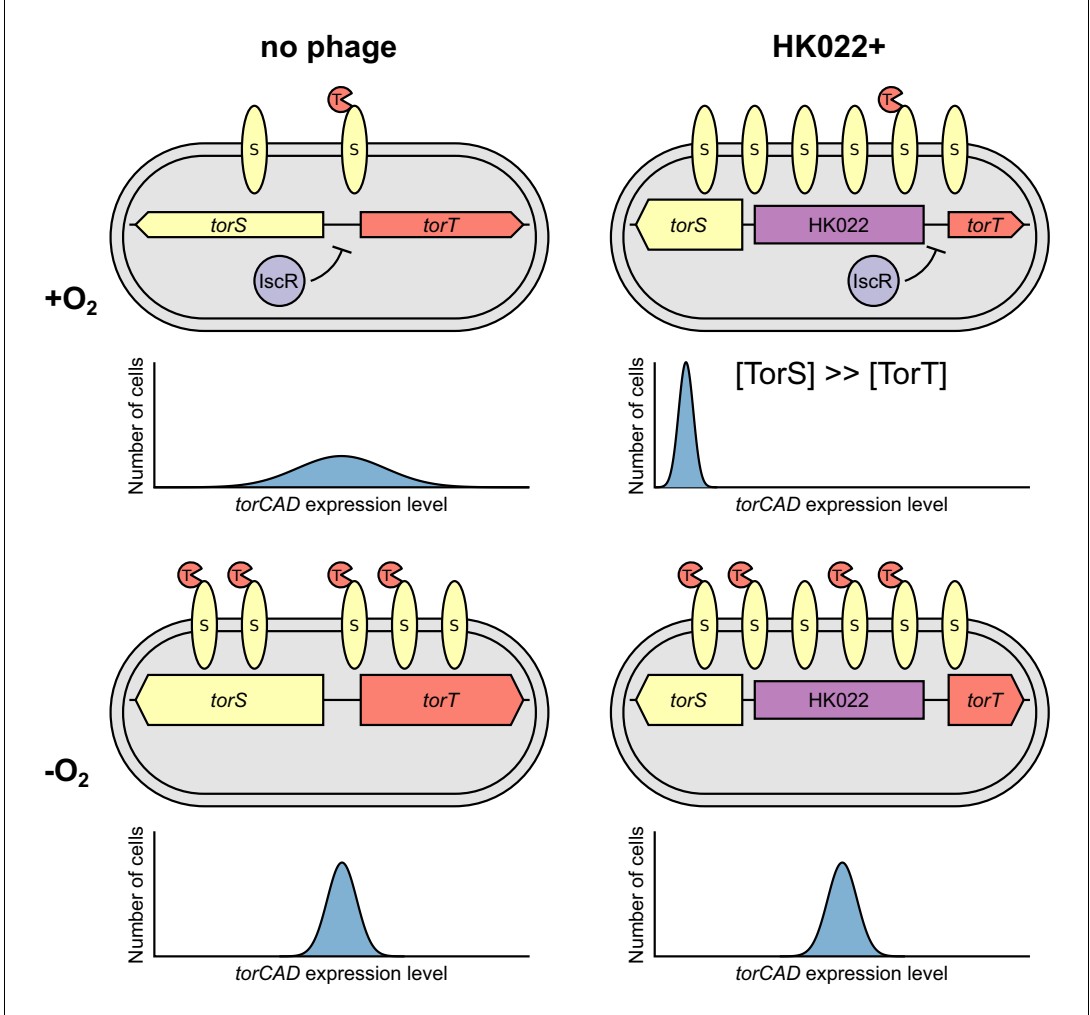

**Figure 5.** Model of how bacteriophage HK022 reprograms the regulation of *torCAD* expression during lysogeny. In cells lacking the HK022 prophage, IscR repression of *torS* and *torT* during aerobic growth leads to very low TorS and TorT abundance. High variability in the ratio of TorS to TorT results in noisy *torCAD* transcription (top left). In the absence of oxygen, IscR repression of *torS* and *torT* is relieved, decreasing variability in the TorS-to-TorT ratio and noise in *torCAD* transcription (bottom left) (*Carey et al., 2018*). In HK022 lysogens, a prophage-encoded promoter drives high *torS* expression. IscR still represses *torT* during aerobic growth, and the resulting excess of TorS relative to TorT shuts down *torCAD* transcription (top right) (see *Figure 1A*). In the absence of oxygen, IscR repression of *torT* is relieved, and *torCAD* transcription is restored (bottom right).
DOI: https://doi.org/10.7554/eLife.49081.016

chromosome (*Figure 3*). These results indicate that an outward-reading transcript originates from within the prophage and reads through *torS*.

We can only speculate on why HK022 shuts off aerobic *torCAD* expression and, consequently, the bet hedging associated with it. We have argued previously that there must be a fitness cost to the expression of *torCAD*, or else its expression would not be regulated (*Carey et al., 2018*). The HK022 prophage may prevent aerobic *torCAD* transcription to alleviate a fitness cost and thereby increase the rate of its own replication. If the primary function of aerobic *torCAD* expression is bet hedging on rapid oxygen depletion, shutting down aerobic expression could be a useful strategy if phages like HK022 primarily lysogenize *E. coli* in environments where TMAO is present but rapid oxygen loss is unlikely to occur. One can conceive of such niches existing within habitats enriched in TMAO, such as the mammalian urinary tract, animal latrines, or the marine environment (especially in association with marine animals).

The HK022 integration site is occupied by a prophage in roughly 5% of fully sequenced *E. coli* strains, and these prophages are found in *E. coli* of diverse origins and phylogenetic groups. This

suggests that HK022 is not an oddity in its integration between *torS* and *torT*. All of the prophages we identified occupying this site have, like HK022, an outward-reading integrase gene as the final identifiable gene before *torS*, and all share the general genomic architecture of phages belonging to the lambda supercluster (*Grose and Casjens, 2014*). (The one partial exception is the prophage from strain STEC299, which maintains the HK022-like integrase oriented towards *torS* but appears to be more closely allied with the GF-2 phage supercluster (*Casjens and Grose, 2016*) than the lambda supercluster). We investigated oxygen-dependent *torCAD* expression in one of these pro-phage-containing strains, the Crohn's disease-associated strain NRG 857C, and found that the *torCAD* expression pattern in this strain is similar to the pattern we observe in MG1655 containing the HK022 prophage. In contrast, a previous study found that two other *E. coli* isolates, Nissle 1917 and HS, which do not have prophages integrated at $attB_{HK022}$, have *torCAD* expression patterns that are like that of wild-type (uninfected) MG1655 (*Roggiani and Goulian, 2015*). These observations suggest that it may be a general capability of the phages that integrate between *torS* and *torT* to alter host regulation of *torCAD* expression. However, despite the similarity of the *torCAD* expression pattern in NRG 857C and HK022-infected MG1655 and the high conservation of the region around $TSS_{HK022}$ in prophage-carrying wild strains, we would not necessarily expect *torCAD* expression in all such strains to have the same behavior: there is considerable variation in overall sequence and gene content among the prophages that occupy the $attB_{HK022}$ site, the host genomes they inhabit, and the habitats from which they were isolated (see *Supplementary file 1*).

Studying the diversity and distribution of prophage-mediated *torCAD* expression could provide insight into the evolutionary advantages for a phage to reconfigure the control of TMAO respiration. Prophage-mediated effects on host physiology remain largely enigmatic, and knowledge is mostly restricted to cases where the effects are readily apparent (as when prophage morons confer an observable phenotype such as toxin production; *Hendrix et al., 2000*). Cases where a prophage directly alters host metabolism have been described infrequently and generally with little mechanistic detail. To our knowledge, the phenomenon described in this study, where a prophage rewires the regulation of a metabolic pathway by modulating the expression of a signaling gene, has not been reported before and may exemplify a general class of mechanisms phages use to control host behavior. Phage infections certainly play a significant role in bacterial community dynamics, and much of our knowledge about the effects of phage infection is centered on lytic infection, horizontal gene transfer, and bacterial pathogenesis. A greater appreciation of the subtler effects of phage infection on host phenotype is a likely platform for developing enhanced understanding of the structure and behavior of microbial communities.

# Materials and methods

**Key resources table**

| Reagent type (species) or resource | Designation | Source or reference | Identifiers | Additional information |
|---|---|---|---|---|
| Gene (*Escherichia coli*) | *torS* | NA | EcoCyc:G6514; UniProt:P39453 | |
| Strain, strain background (*Escherichia virus HK022*) | HK022 | PMID: 4569213 | RefSeq:NC_002166 | Dr. Max E. Gottesman (Columbia University) |
| Strain, strain background (*E. coli*) | DFE12 | this paper | | MG1655 $attB_\lambda$:: (*cat* $P_{torCAD}$-*yfp*) *ompA*-*cfp* $(HK022)_n$ |
| Strain, strain background (*E. coli*) | DFE34 | this paper | | NRG 857C Δ*lacIZY*:: $P_{torCAD}$-*yfp*-FRT-*kan*-FRT |
| Strain, strain background (*E. coli*) | JNC151 | this paper | | MG1655 $(HK022)_n$ |

*Continued on next page*

*Continued*

| Reagent type (species) or resource | Designation | Source or reference | Identifiers | Additional information |
|---|---|---|---|---|
| Strain, strain background (*E. coli*) | JNC163 | PMID: 29502970 | | MG1655 Δ*lacZYA*::FRT-*cat*-FRT *torT-lacZ*-FRT-*kan*-FRT Δ*torR* |
| Strain, strain background (*E. coli*) | JNC166 | PMID: 29502970 | | MG1655 Δ*lacZYA*::FRT *torS-lacZ*-FRT-*kan*-FRT |
| Strain, strain background (*E. coli*) | JNC168 | this paper | | MG1655 Δ*lacZYA*::FRT-*cat*-FRT (HK022)$_n$ *torT-lacZ*-FRT-*kan*-FRT Δ*torR* |
| Strain, strain background (*E. coli*) | JNC169 | this paper | | MG1655 Δ*lacZYA*::FRT *torS-lacZ*-FRT-*kan*-FRT (HK022)$_n$ |
| Strain, strain background (*E. coli*) | JNC173 | this paper | | MG1655 Δ*fhuA*::FRT-*kan*-FRT *attB*$_\lambda$::(*cat* P$_{torCAD}$-*yfp*) *ompA-cfp* (HK022)$_n$ |
| Strain, strain background (*E. coli*) | JNC174 | this paper | | MG1655 Δ*fhuA*::FRT-*kan*-FRT *attB*$_\lambda$::(*cat* P$_{torCAD}$-*yfp*) Δ*xylAFG*::P$_{tetA}$-*mcherry*-FRT |
| Strain, strain background (*E. coli*) | JNC175 | this paper | | MG1655 Δ*lacZYA*::FRT *torS-lacZ*-FRT-*kan*-FRT (HK022)$_n$ *attL*$_{HK022}$::Ω |
| Strain, strain background (*E. coli*) | MG1655 | Coli Genetic Stock Center | CGSC:7740; RefSeq: NC_000913 | |
| Strain, strain background (*E. coli*) | MMR8 | PMID: 25825431 | | MG1655 *attB*$_\lambda$::(*cat* P$_{torCAD}$-*yfp*) *ompA-cfp* |
| Strain, strain background (*E. coli*) | NRG 857C | PMID: 21108814 | RefSeq: NC_017634 | Dr. Alfredo G. Torres (UTMB) |
| Strain, strain background (*E. coli*) | PK13196 | this paper | | MG1655 *lacZ*::*kan*-P$_{torS}$-(GTG)*lacZ* Δ*iscR*::FRT |
| Strain, strain background (*E. coli*) | PK13199 | this paper | | MG1655 *lacZ*::*kan*-P$_{torS}$-(ATG)*lacZ* Δ*iscR*::FRT |
| Recombinant DNA reagent | pDSW206 | PMID: 9882665 | | ori(pBR322) *lacI*$^q$ *amp* P$_{trc}$ attenuated promoter. Dr. Jon Beckwith (Harvard University) |
| Recombinant DNA reagent | pMR26 | PMID: 25825431 | | pDSW206 *torT* |
| Recombinant DNA reagent | pPK7179 | PMID: 15659690 | | ori(pBR322) ter(*spf*) *amp* RNA-1 |
| Recombinant DNA reagent | pPK12669 | this paper | | pPK7179 with −152 to +28 bp relative to the *torS* ATG start codon from MG1655 in XhoI/BamHI sites |

*Continued on next page*

*Continued*

| Reagent type (species) or resource | Designation | Source or reference | Identifiers | Additional information |
|---|---|---|---|---|
| Recombinant DNA reagent | pPK13256 | this paper | | pPK7179 with −231 to +28 bp relative to the *torS* ATG start codon from JNC151 in XhoI/BamHI sites |
| Sequence-based reagent | native *torS* | this paper | | $^{32}$P-labeled DNA oligonucleotide: 5′-TTAACAGCGCCATCAG-3′ |
| Sequence-based reagent | HK022/*torS* | this paper | | $^{32}$P-labeled DNA oligonucleotide: 5′-GGGTCAGGGTTAAATTCACGG-3′ |
| Peptide, recombinant protein | *E. coli* $\sigma^{70}$ RNA polymerase holoenzyme | New England Biolabs | NEB:M0551S | |
| Commercial assay or kit | HiSpeed Plasmid Maxi Kit | Qiagen | Qiagen:12662 | |
| Commercial assay or kit | MMLV Reverse Transcriptase 1st-Strand cDNA Synthesis Kit | Lucigen | Lucigen: MM070150 | |
| Commercial assay or kit | Sequenase Version 2.0 DNA Sequencing Kit | USB | USB:70770 | |
| Software, algorithm | BLAST | PMID: 23609542 | RRID:SCR_004870 | |
| Software, algorithm | ClermonTyping | PMID: 29916797 | | v. 1.4.0 |
| Software, algorithm | ggridges | Comprehensive R Archive Network | RRID: SCR_003005 | v. 0.5.0 |
| Software, algorithm | Mauve | PMID: 20593022 | RRID: SCR_012852 | v. 2015-02-25 |
| Software, algorithm | MUSCLE | PMID: 15034147 | | v. 3.8.1551 |
| Software, algorithm | R | R Foundation for Statistical Computing | RRID: SCR_001905 | v. 3.4.4 |
| Software, algorithm | SnapGene | GSL Biotech | RRID: SCR_015052 | v. 5.0b3 |

## Bacterial growth media and conditions

Media and growth conditions were as described in *Carey et al. (2018)* except that minimal A glucose medium was supplemented with 0.1% casamino acids and 10 mM TMAO for all experiments. Antibiotics were added to media at the following concentrations unless otherwise indicated: streptomycin, 250 µg/mL; ampicillin, 50 µg/mL; kanamycin, 25 µg/mL; and spectinomycin 20 µg/mL.

## Strain construction

Lists of all strains and plasmids used in this study are provided in *Supplementary file 3* and *Supplementary file 4*, respectively. HK022 was a generous gift from M.E. Gottesman (Columbia University). P1*vir* transductions were performed as in *Miller (1992)* to create strains JNC173 (JW0146 × DFE12) and JNC174 (JW0146 × MMR65). HK022 lysogens were generated using a method adapted from protocols for making λ lysogens (*Silhavy et al., 1984*) and for making mycobacteriaphage lysogens (*Sarkis and Hatfull, 1998*). Briefly, the strain to be lysogenized was grown to saturation in LB and harvested by centrifugation. Cells were resuspended at 2× concentration in 10 mM MgSO$_4$, and 100 µL of the suspension was added to 3 mL molten LB top agar at 45°C. The

top agar was mixed, layered onto an LB agar plate prewarmed to 42°C, and allowed to solidify. HK022 lysate (50 µL) was spotted onto the top agar and allowed to dry, and the plate was incubated at 37°C overnight. On the following day, an LB plate was spread with 100 µL HK022 lysate and allowed to dry. Selection for lysogens was carried out by streaking from the turbid zone of lysis formed on the top agar plate onto the HK022-spread LB plate and incubating at 37°C overnight. HK022-resistant colonies were patched onto LB agar, and the same colonies were tested for lysogeny by patching onto a top agar lawn containing an HK022-sensitive strain (MG1655). After overnight incubation at 37°C, candidate lysogens that produced a zone of lysis around the area of the patch (from spontaneous phage release) were nonselectively purified by streaking for single colonies from the LB plate patches and incubating at 37°C overnight. The entire patch test procedure was then repeated using the purified colonies. Candidate lysogen colonies that still produced a zone of lysis around the patch after purification were tested for the presence of the HK022 prophage by PCR. Strains produced by this method were JNC151 (HK022 lysogen of MG1655), DFE12 (HK022 lysogen of MMR8), JNC168 (HK022 lysogen of JNC163), and JNC169 (HK022 lysogen of JNC166). These strains were assayed for tandem polylysogeny by PCR essentially as in *Powell et al. (1994)* using primers HK022-P1 (5'-GGAATCAATGCCTGAGTG-3'), HK022-P2 (5'-GCTGATACACTACAG-CAATG-3'), HK022-P3 (5'-GACAGGAGCTTGTTGACTAA-3'), and HK022-P4 (5'-GGCATCAACAG-CACATTC-3'). All appeared to be tandem polylysogens (denoted $(HK022)_n$ in the Key Resources Table and *Supplementary file 3*, following the convention of *King et al., 2000*), although the possible presence of contaminating virion DNA in the PCR template could not be ruled out.

The Ω element strain JNC175 was constructed by recombineering (*Datsenko and Wanner, 2000*). The Ω element was amplified by PCR from pJB31 using primers LRpJB31_JNC169U1 (5'-CAGAGTCTTCGGGTCAGGGTTAAATTCACGGTCGGTGCACTTTAGGTGAAATCCCGAATGTGCAG TTAAC-3') and LRpJB31_JNC169L1 (5'-TACTTACATTAATTTACTGATAATTAAAGAGATTTTAAATA TACAACTTAGGCGCTGAAAGAAACCGCAA-3'). The PCR product was digested with DpnI and purified before electroporation into JNC169 carrying helper plasmid pKD46. Cultures were spread on LB agar plates containing streptomycin (20 µg/mL), spectinomycin (20 µg/mL), and kanamycin (25 µg/mL) to select for integration of the Ω element and maintenance of the *torS-lacZ* fusion. The strain was cured of pKD46, and correct integration of the Ω element into $attL_{HK022}$ was verified by sequencing. The $attL_{HK022}::\Omega$ construct was transduced into a clean JNC169 background to create JNC175.

To construct strains containing chromosomally encoded $P_{torS}$-*lacZ* translational fusions, a DNA fragment encompassing -152 to +28 bp relative to the *torS* ATG start codon (as determined in this study) was PCR amplified from the MG1655 chromosome and cloned into the XhoI and BamHI sites of pPK7035 upstream of *lacZ'*, creating pPK12792. pPK12792 served as a template for site directed mutagenesis in which bases downstream of predicted *torS* start codons (GTG or ATG) through the native *lacZ* start codon were deleted to create plasmids harboring the translational fusion constructs *kan*-$P_{torS}$-(GTG)*lacZ'* (pPK13169) and *kan*-$P_{torS}$-(ATG)*lacZ'* (pPK13171). pPK13169 and pPK13171 were used as templates for PCR amplification of the translational fusion constructs using primers with homology to the native $P_{lac}$ region. The amplicons were electroporated into PK12556, and kanamycin resistance was used to select for integrants. The *kan*-$P_{torS}$-(GTG)*lacZ* and *kan*-$P_{torS}$-(ATG) *lacZ* constructs were then moved into PK4854 using P1*vir* transduction to create PK13196 and PK13199, respectively.

For in vitro transcription and primer extension assays, promoter regions were cloned into the XhoI and BamHI sites of pPK7179. For native $P_{torS}$, the aforementioned DNA fragment encompassing -152 to +28 bp relative to the *torS* ATG start codon was used, generating pPK12669. To identify the HK022-derived promoter driving *torS* expression, a DNA fragment encompassing -231 to +28 bp relative to the *torS* start codon was PCR amplified from the chromosome of JNC151 and cloned into pPK7179, generating pPK13256.

The $P_{torCAD}$-*yfp* reporter was introduced into NRG 857C by conjugation. As *E. coli* K-12 and *E. coli* NRG 857C are not closely related, synteny of their chromosomes was confirmed by genomic alignment using Mauve (*Darling et al., 2004*; *Darling et al., 2010*) before proceeding. The $P_{torCAD}$-*yfp* reporter was first moved from strain MMR129 into the Hfr strain SASX41B by P1*vir* transduction, creating DFE33. DFE33 retains the *hemA41* allele of SASX41B and is therefore a δ-aminolevulinic acid auxotroph. DFE33 was mated with NRG 857C by superimposed patching of one colony of each strain onto an LB agar plate supplemented with 25 µg/mL δ-aminolevulinic acid. The plate

was incubated overnight at 37°C, and on the following day bacteria growing in the patch area were streaked for single colonies onto LB agar supplemented with kanamycin and lacking δ-aminolevulinic acid. This selective media permitted growth of cells that had received the P$_{torCAD}$-*yfp* reporter, which is linked to a kanamycin resistance gene, but had not received the *hemA41* allele: as the *hemA* locus is proximal to the *tor* locus in the direction of conjugative transfer, growth without δ-aminolevulinic acid indicated retention of the NRG 857C *tor* genes. Colonies were purified nonselectively on LB agar, and the resulting strain was named DFE34. The *metB1* allele of DFE33, which confers methionine auxotrophy, was also verified not to have been transferred to DFE34 by confirming that DFE34 could grow on minimal glucose medium without amino acid supplementation. The *metB* locus is proximal to the *iscR* locus (and the *hemA* and *tor* loci) in the direction of conjugative transfer, so growth on minimal medium without amino acid supplementation indicated that DFE34 retained the NRG 857C *iscR* allele. Based on the genetic markers analyzed, the maximum amount of NRG 857C genomic sequence that could have been replaced by K-12 sequence during the construction of DFE34 is 1.1 Mbp.

## Phase contrast and fluorescence microscopy

Microscopy was performed as described in *Carey et al. (2018)* except that cultures were grown to OD$_{600}$ = 0.1–0.4 before being put on ice. Cultures were chilled on ice for 30 min at the time of streptomycin addition and then aerated on a roller drum at 37°C for 2 hr before being held at 4°C overnight. Imaging was performed the next day with no additional aeration beforehand. *Figures 1* and *4* were generated using the R package ggridges (*R Development Core Team, 2018*; *Wickham, 2009*; *Wilke, 2018*). The density curves were generated using a Gaussian kernel function with the bandwidth selected by applying Silverman's rule of thumb (*Silverman, 1986*) to the entire data set.

## Aerobic-to-anaerobic transition microscopy

Aerobic-to-anaerobic transition microscopy was performed as described in *Carey et al. (2018)* except that no Δ*torC* control strain was included.

## β-Galactosidase assays

β-Galactosidase assays were performed as in *Carey et al. (2018)* except that cultures were grown to OD$_{600}$ = 0.1–0.5 before harvesting. For the β-galactosidase assays using the P$_{torS}$-*lacZ* translational fusion strains PK13196 and PK13199, chloramphenicol was added to the cultures at a final concentration of 20 μg/mL before placing on ice.

## In vitro transcription assays

Following purification of pPK12669 and pPK13256 with a HiSpeed Plasmid Maxi Kit (Qiagen), 2 nM supercoiled plasmid was incubated with 5 μCi of [α-$^{32}$P]UTP, 50 μM unlabeled UTP, and 500 μM final concentrations each of ATP, CTP, and GTP for 5 min at 37°C in 40 mM Tris (pH 7.9), 30 mM KCl, 100 μg/mL bovine serum albumin, 1 mM dithiothreitol, and 10 mM MgCl$_2$. *E. coli* σ$^{70}$ RNA polymerase holoenzyme (50 nM) was added, and each reaction (20 μl total volume) was terminated after 10 min by adding 10 μL 95% (vol/vol) formamide, 20 mM EDTA, 0.05% (wt/vol) bromophenol blue, and 0.05% (wt/vol) xylene cyanol FF. After the mixture was heated to 90°C for 30 s, 5 μl was loaded onto an 8% polyacrylamide-7 M urea gel (0.5× TBE) and run at 1400 V for 3 hr. The gel was then dried and exposed to a PhosphorImager screen.

## Primer extension assays

RNA was synthesized using the same protocol as for the in vitro transcription assays, with the exception that UTP was unlabeled. After phenol extraction and ethanol precipitation, 5 μg RNA was hybridized with a $^{32}$P-labeled primer ('native *torS*' or 'HK022/*torS*') by heating at 95°C for 5 min followed by slow cooling for 1 hr. Primer extension with the MMLV Reverse Transcriptase 1st-Strand cDNA Synthesis Kit (Lucigen) was carried out according to the manufacturer's instructions. Sequencing reactions using the same primer from the primer extension assays were performed using the Sequenase Version 2.0 DNA Sequencing Kit (USB).

## Sequence analysis

Phylogenetic group assignments of the prophage-carrying strains listed in *Supplementary file 1* were made as described in *Clermont et al. (2013)* using the ClermonTyping web tool (*Beghain et al., 2018*) and strain sequences available from NCBI. Isolation source was identified from information in the NCBI sequence entry or linked BioSample entry (*Barrett et al., 2012*). The most similar phage was identified using PHASTER (*Arndt et al., 2016*) and is the fully sequenced phage with the highest overall protein sequence similarity to the query prophage. Prophage completeness was assessed using PHASTER. Multiple sequence alignment was performed using MUSCLE (*Edgar, 2004*) as implemented in SnapGene (*SnapGene, 2019*).

## Acknowledgements

This work was funded by grants from the National Institute of General Medical Sciences of the NIH: R01-GM080279 to MG and R01-GM115894 to PJK.

## Additional information

### Funding

| Funder | Grant reference number | Author |
| --- | --- | --- |
| National Institutes of Health | R01-GM080279 | Mark Goulian |
| National Institutes of Health | R01-GM115894 | Patricia Kiley |

The funders had no role in study design, data collection and interpretation, or the decision to submit the work for publication.

### Author contributions

Jeffrey N Carey, Conceptualization, Investigation, Writing—original draft, Writing—review and editing; Erin L Mettert, Daniel R Fishman-Engel, Manuela Roggiani, Conceptualization, Investigation, Writing—review and editing; Patricia J Kiley, Conceptualization, Supervision, Funding acquisition, Investigation, Writing—review and editing; Mark Goulian, Conceptualization, Supervision, Funding acquisition, Investigation, Writing—original draft, Writing—review and editing

### Author ORCIDs

Jeffrey N Carey (ID) https://orcid.org/0000-0002-5105-0265
Mark Goulian (ID) https://orcid.org/0000-0003-0076-023X

### Decision letter and Author response

Decision letter https://doi.org/10.7554/eLife.49081.024
Author response https://doi.org/10.7554/eLife.49081.025

## Additional files

### Supplementary files

• Source data 1. FASTA sequence alignment file. Source data for *Supplementary file 2*.
DOI: https://doi.org/10.7554/eLife.49081.017

• Supplementary file 1. Fully sequenced *E. coli* isolates carrying prophages at the HK022 integration site.
DOI: https://doi.org/10.7554/eLife.49081.018

• Supplementary file 2. Sequence alignment of the genomic region around the prophage-encoded *torS* transcription start site from all fully sequenced *E. coli* strains carrying a prophage at the HK022 integration site.
DOI: https://doi.org/10.7554/eLife.49081.019

• Supplementary file 3. Strains used in this study.

DOI: https://doi.org/10.7554/eLife.49081.020

• Supplementary file 4. Plasmids used in this study.
DOI: https://doi.org/10.7554/eLife.49081.021

• Transparent reporting form DOI: https://doi.org/10.7554/eLife.49081.022

## Data availability

All data generated or analysed during this study are included in the manuscript and supporting files.

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
