## [Decision Letter]

Thank you for submitting your article "Phage integration alters the respiratory strategy of its host" for consideration by *eLife*. Your article has been reviewed by three peer reviewers, one of whom is a member of our Board of Reviewing Editors, and the evaluation has been overseen by Gisela Storz as the Senior Editor. The reviewers have opted to remain anonymous.The reviewers have discussed the reviews with one another and the Reviewing Editor has drafted this decision to help you prepare a revised submission.

This paper presents evidence that integration of the temperate phage HK022 into its primary integration site between *torS* and *torT* leads to significant changes in the respiratory strategy of *E. coli*. In prior work the authors had found that the mean response of cells /- O_2_ was unchanged but the variance changed. Now, they find that with HK022 integrated, cells exhibit a clear change in mean response with the TorS/R pathway on in anaerobic conditions and off in aerobic conditions. Follow up studies demonstrate that this difference likely stems from transcription of *torS* from a phage promoter. Although the authors do not specifically demonstrate how this metabolic change enhances host or phage fitness, it seems likely that there is some advantage to this prophage insertion under some conditions. Given all the possible places that a phage could integrate, the fact that HK022 lands in such a sensitive position and has a promoter to compensate for loss of the *torS* promoter indicates that there is an adaptive advantage to this phenomenon. Overall, the reviewers were enthusiastic about the paper and agreed that it warrants publication in *eLife*. There were, however, a few suggestions, collated below, for improvement that should be addressed in a revision.

1) I wondered about the role of non-specific phosphodonors for TorR, e.g. acetyl-phosphate, which can sometimes drive some RR phosphorylation, especially when a cognate kinase is not present or highly expressed. Is it possible that in the no phage, O_2_ condition there is higher variability in acetyl-phosphate, leading to the variability and higher response relative to the phage, O_2_ condition? E.g. maybe presence of the phage somehow affects central metabolism to reduce acetyl-phosphate levels, leading to a tighter OFF state in the phage, O_2_ condition? I think the experiments presented indicate that higher TorS levels are a more likely explanation for what's going on, but maybe there's a contribution from a change in other phosphodonors?

2) In the Introduction, the authors should provide a small amount more background into the importance of the *torCAD* system. For example, what is TMAO and why is it important in the absence of oxygen. In the Discussion, this increased background might allow the authors to give a somewhat more detailed idea of what advantage the HK022 prophage could provide. For example, what kind of conditions produce high TMAO concentrations.

3) In discussing the observation that different phages integrate at the same site as HK022, the authors should emphasize that these phages are definitely distinct phages from HK022. For example, some of the strains listed in Table 1 have P22-like phages integrated at the HK022 *att*. These phages have a completely different type of tail. Do these prophages all have the same integrase? Do any of them have different *att* sites but still integrate in the same region. This would strongly support the adaptive benefit of integrating in the *torS* region.

4) While I could not find examples in the literature of the type of prophage-induced transcriptional re-wiring described here, there are examples of the insertion position of a prophage playing key roles in bacterial physiology. One example is the skn prophage in *B. subtilis*. This and some other examples can be found in recent reviews (e.g. Feiner et al., 2015). Some of these examples could be mentioned to put the current work into context.

5) No mention of spontaneous induction of the HK022 prophage is made. I submit that bet hedging might be going on at a lower level if there is a reasonable level of spontaneous excision going on. It would be easy to estimate the frequency of excision events using Q-PCR and properly selected primer pairs.

6) The biggest omission is some sort of tabulation of the prophage promoters that should be found in the multiple wild type *E. coli* strains that have the prophages. They (laboriously) tested 1 wt prophage strain and found the same abrogation of the stochastic regulation going on. Surely we should see whether the promoter is conserved or whether other promoters are there?

7) One experimental concern is that the authors do not provide in vivo evidence for the activity of the prophage promoter. There are enough weird things that bacteria do to prophage sequences that I would like to see a Northern or some such from in vivo situations.

---

## [Author Response]

1) I wondered about the role of non-specific phosphodonors for TorR, e.g. acetyl-phosphate, which can sometimes drive some RR phosphorylation, especially when a cognate kinase is not present or highly expressed. Is it possible that in the no phage, O_2_ condition there is higher variability in acetyl-phosphate, leading to the variability and higher response relative to the phage, O_2_ condition? E.g. maybe presence of the phage somehow affects central metabolism to reduce acetyl-phosphate levels, leading to a tighter OFF state in the phage, O_2_ condition? I think the experiments presented indicate that higher TorS levels are a more likely explanation for what's going on, but maybe there's a contribution from a change in other phosphodonors?

Available data suggests phosphorylation of TorR is exclusively dependent on TorS kinase activity, as *torS* strains show no expression of TorR-dependent genes (Jourlin et al.. 1996). Based on this evidence, which was acquired under the same culture conditions as in the present work, nonspecific phosphodonors such as acetyl phosphate do not appear to play a significant role in phosphorylating TorR and activating *torCAD* transcription.

2) In the Introduction, the authors should provide a small amount more background into the importance of the torCAD system. For example, what is TMAO and why is it important in the absence of oxygen. In the Discussion, this increased background might allow the authors to give a somewhat more detailed idea of what advantage the HK022 prophage could provide. For example, what kind of conditions produce high TMAO concentrations.

Additional background on *torCAD* and TMAO has been added to the Introduction, and some further speculation on the environments in which lysogeny could be advantageous has been added to the Discussion.

3) In discussing the observation that different phages integrate at the same site as HK022, the authors should emphasize that these phages are definitely distinct phages from HK022. For example, some of the strains listed in Table 1 have P22-like phages integrated at the HK022 att. These phages have a completely different type of tail. Do these prophages all have the same integrase? Do any of them have different att sites but still integrate in the same region. This would strongly support the adaptive benefit of integrating in the torS region.

We have extended this table (now Supplementary file 1) to include new *E. coli* genome sequences available from NCBI since the initial submission. In the 45 genomes containing a prophage upstream of *torS*, all have the prophage integrated at a site identical to *attB*_HK022_ (with two SNPs observed in the core region of the *att* sequence—see the new Supplementary file 2 for a multiple sequence alignment). Additionally, the integrase of each of these prophages shares the highest conservation with HK022 integrase (out of all the roughly 187,000 viral sequences searched by PHASTER). We found the reviewer’s comment about the different tail types of some of the phages interesting and have accordingly added a column to the table listing the most similar known phage (again as identified by PHASTER) for the benefit of the intent reader.

4) While I could not find examples in the literature of the type of prophage-induced transcriptional re-wiring described here, there are examples of the insertion position of a prophage playing key roles in bacterial physiology. One example is the skn prophage in B. subtilis. This and some other examples can be found in recent reviews (e.g. Feiner et al., 2015). Some of these examples could be mentioned to put the current work into context.

Thank you for directing us to this interesting reference. In this revision we have included additional examples of integration site-dependent gene regulation by prophages and citation of the Feiner et al. article.

5) No mention of spontaneous induction of the HK022 prophage is made. I submit that bet hedging might be going on at a lower level if there is a reasonable level of spontaneous excision going on. It would be easy to estimate the frequency of excision events using Q-PCR and properly selected primer pairs.

We grew an HK022 lysogen under culture conditions identical to those used in the bet-hedging experiments and performed PCR across the HK022 *attB* site to assay for prophage excision events. We did find that HK022 excision occurs, but by PCR we do not know how to differentiate between excision events leading toward virion production/cell lysis and excision events not leading to cell death (e.g. as in active lysogeny described in the Feiner et al. reference above). However, if spontaneous HK022 excision were providing another layer of bet hedging regulation, we would expect to see a subpopulation of cells in the aerobic HK022 lysogen culture with high *torCAD* expression, and we do not see any such cells (see Figure 1). This suggests that such excision events, if they occur, are rare.

6) The biggest omission is some sort of tabulation of the prophage promoters that should be found in the multiple wild type E. coli strains that have the prophages. They (laboriously) tested 1 wt prophage strain and found the same abrogation of the stochastic regulation going on. Surely we should see whether the promoter is conserved or whether other promoters are there?

Unfortunately, we do not know the precise location of the prophage-encoded promoter, only the associated transcription start site. However, an alignment of the region upstream of TSS_HK022_(*torS*)—the transcription start site that we identified—shows that this sequence is highly conserved across the 45 prophage-containing genomes. We have provided this alignment in the new Supplementary file 2.

*7) One experimental concern is that the authors do not provide* in vivo *evidence for the activity of the prophage promoter. There are enough weird things that bacteria do to prophage sequences that I would like to see a Northern or some such from* in vivo *situations.*

We were unable to detect transcripts expressed from the prophage promoter in vivo in attempts to perform TSS mapping by primer extension using endogenous mRNA. Although disappointing, this was unsurprising considering the exceptionally low transcription rate of wild-type *torS* in vivo (based on our measurements with transcriptional reporters and the low protein expression levels reported in, for example, Taniguchi et al., 2010 and Li et al., 2014). However, we feel that our measurements of *torS* transcription with and without a synthetic terminator integrated between the prophage and *torS* coding sequence (Figure 3B) provide strong evidence that the prophage promoter is active in vivo.